# Backward Walking as a Marker of Mobility and Disability in Multiple Sclerosis: A Cross-Sectional Analysis

**DOI:** 10.3390/diagnostics15070936

**Published:** 2025-04-06

**Authors:** Meral Seferoğlu, Abdulkadir Tunç, Ali Özhan Sıvacı, Samed Öncel, Tuğba Düztaban, Hamide Dikilitaş, Abdul Samed Görgül, Muhammed Furkan Öztürkci

**Affiliations:** 1Department of Neurology, University of Health Sciences, Bursa Yüksek Ihtisas Training and Research Hospital, Bursa 16310, Turkey; meralbozseferoglu@gmail.com (M.S.);; 2Department of Neurology, Faculty of Medicine, Sakarya University, Sakarya 54100, Turkey; 3Department of Neurology, Sakarya Education and Research Hospital, Sakarya 54100, Turkey

**Keywords:** multiple sclerosis, mobility limitation, gait disorders, walking speed, cognitive dysfunction

## Abstract

**Background:** Mobility impairments in multiple sclerosis (MS) significantly affect quality of life. This study evaluated the clinical utility and sensitivity of the Backward Timed 25-Foot Walk Test (B-T25FW) and its associations with key clinical measures in MS. **Methods**: A cross-sectional study was conducted with 129 ambulatory patients with MS from two centers. Disability (Expanded Disability Status Scale, EDSS), cognition (Symbol Digit Modalities Test, SDMT), manual dexterity (Nine-Hole Peg Test, 9HPT), fatigue, and forward and backward walking were assessed. Correlation and receiver operating characteristic (ROC) analyses were performed. **Results**: The participants included in the study were 76% female, with a mean age of 38 years, and the majority were diagnosed with relapsing–remitting MS (86.8%). Backward and forward walking times significantly correlated with key clinical measures, including the EDSS, SDMT, and 9HPT. Backward walking times showed moderate correlations with EDSS (r = 0.469) and weaker but significant correlations with 9HPT (r = 0.452) and disease duration (r = 0.245). Both walking tests were negatively correlated with SDMT scores. For prognostic purposes, forward walking exhibited slightly higher predictive power compared to backward walking. **Conclusions**: The B-T25FW is a clinically relevant, practical, and sensitive tool for assessing mobility impairments in individuals with MS. Its integration into clinical practice could complement forward walking assessments, enhancing disease monitoring and guiding interventions. Future research should validate its longitudinal utility.

## 1. Introduction

Multiple sclerosis (MS) is a progressive neurological disorder characterized by inflammation, demyelination, and axonal degeneration, leading to motor, cognitive, and sensory impairments [1]. Among these deficits, mobility impairments are some of the most disabling symptoms, with over 70% of patients with MS identifying walking difficulties as their most significant challenge [2]. Walking deficits not only increase the risk of falls but also contribute to activity curtailment, social isolation, and reduced quality of life [3,4]. The Timed 25-Foot Walk Test (T25FW) has become a standard tool for assessing ambulatory function in MS and is a key component of the Multiple Sclerosis Functional Composite (MSFC) [5]. By measuring the time required to walk 25 feet at the maximum safe speed, the T25FW provides a sensitive assessment of walking ability, correlating strongly with the Expanded Disability Status Scale (EDSS) and demonstrating superior sensitivity in detecting disease progression [6]. While forward walking forms the basis of the T25FW, backward walking requires greater cognitive and proprioceptive engagement and has been shown to differentiate fallers from non-fallers in elderly populations and various neurodegenerative conditions [7]. In MS, backward walking deficits have been associated with increased disability and higher fall risk, often exceeding the predictive value of forward walking metrics [8,9]. Specifically, the backward version of the T25FW captures aspects of postural control and mobility impairment unique to patients with MS [7]. Recent studies have further highlighted the importance of backward walking as a marker for functional mobility and fall risk. For example, the 3 m backward walk test demonstrates excellent test–retest reliability and correlates with measures of disease severity, balance, and activity levels [10]. However, despite its potential, backward walking remains underexplored in MS, with gaps in understanding its utility for predicting falls and disease progression over time.

This study aimed to assess the clinical utility and sensitivity of the Backward Timed 25-Foot Walk Test (B-T25FW) by examining its correlations with disability status, cognition, and upper-limb function in people with MS.

## 2. Methods

This study was conducted across two premier MS clinics in Turkey, collecting data from September 2024 to January 2025. Ethical approval was secured from the Institutional Review Board (approval number: 2024-TBEK 2024/10-02), and written informed consent was obtained from all participants in line with the Declaration of Helsinki.

### 2.1. Participants

A power analysis was conducted before starting the study, and the appropriate sample size was determined. A sample size of 57 achieves 81% power to detect a difference of −0.2 between the null hypothesis mean of 1.21 and the alternative hypothesis mean of 1.45 with an estimated standard deviation of 0,6 and a significance level (alpha) of 0.05 using a two-sided one-sample *t*-test. With the final sample size of 129, the one-sample *t*-test’s achieved statistical power is on the order of 99%. Participants were required to have a confirmed diagnosis of MS according to the 2017 McDonald criteria [11]. The inclusion criteria stipulated that participants be at least 18 years old, ambulatory (with or without assistive devices), and capable of completing study assessments. Exclusion criteria included recent MS exacerbations (within 30 days), comorbid neurological or orthopedic disorders affecting mobility, and corticosteroid use in the prior month. Individuals unable to perform the walking tests or with incomplete datasets were also excluded. Initially, a pool of approximately 1400 patients under clinical follow-up was considered for the study. During the data collection period, 350 patients attended routine follow-ups and were evaluated for eligibility. Of these, 45 patients were excluded due to comorbid neurological or orthopedic conditions that could significantly impact walking performance, and 30 patients were excluded due to recent MS exacerbations or corticosteroid use. Additionally, 65 patients declined to participate in the study despite meeting the inclusion criteria. An additional 81 patients were excluded due to incomplete datasets or inability to perform the walking assessments. This process resulted in a final cohort of 129 participants who met all eligibility criteria and completed the study protocol.

### 2.2. Procedures

Demographic, clinical, and functional metrics were systematically collected. Demographic data encompassed age, sex, and body mass index (BMI). Clinical assessments included disease duration, age at first MS attack, and disability evaluation via the EDSS. Disease-modifying drugs (DMDs) related to MS were classified as low-efficacy drugs (fumarates, glatiramer acetate, interferons, and teriflunomide) and high-efficacy drugs (cladribine, fingolimod, alemtuzumab, natalizumab, and ocrelizumab), consistent with the previous literature [12]. Cognitive function was assessed using the Symbol Digit Modalities Test (SDMT), a validated tool for measuring information processing speed in patients with MS [13]. Manual dexterity was evaluated with the Nine-Hole Peg Test (9HPT), a standard measure for upper limb functionality in MS [14]. Fatigue was quantified using the Fatigue Severity Scale (FSS) and a visual analog scale (VAS), both of which are widely used and validated in MS populations [15]. Walking performance was evaluated using the T25FW for both forward and backward walking. Each participant completed two trials at their maximum safe walking speed for each direction. The T25FW is a widely recognized and reliable measure of ambulatory function in MS [5]. Tests were conducted on a flat, clear surface with a two-meter buffer zone before the start and after the finish lines to accommodate acceleration and deceleration. Participants were instructed to maintain a forward gaze during the trials to minimize visual distractions. To ensure safety, participants wore gait belts, and trained staff closely supervised all tests.

### 2.3. Data Collection

Walking times were recorded with a stopwatch, and the mean value of the two trials was used for analysis. Clinical measures, including the EDSS, SDMT, NHPT, and fatigue scores, were collected alongside walking performance data to examine their interrelationships. Quality of life metrics were assessed using an MS-specific scale by asking patients to rate the impact of MS on their quality of life during the test, using a scale from 0% (no impact) to 100% (completely impacts my entire life). The backward variant of the T25FW (B-T25FW) adhered to similar protocols as the forward test, requiring participants to walk backward at a controlled pace.

A 10-point scoring system was developed to assess the prognosis of patients with MS, focusing on factors identified in the literature as being associated with poor outcomes [16,17]. Similar approaches to identifying prognostic factors have been explored in the literature, highlighting the significance of demographic, clinical, and radiological characteristics in predicting MS progression and disability accumulation [17]. The participants were divided into two groups based on favorable and unfavorable prognostic features. The parameters for unfavorable prognosis included age at MS onset (>40 years), sex (male), smoking status (current or past smoker), initial presentation (poly-symptomatic), comorbidities (diabetes, prediabetes, hypertension, hypercholesterolemia, obesity), disability at diagnosis (EDSS ≥ 3), baseline T2 lesion count (≥9 lesions), infratentorial lesions, gadolinium-enhancing lesions, and cervical spinal cord lesions (on MRI). Each parameter contributing to the score was assigned 1 point, with a total score above 6 indicating a poor prognosis. Patients with more than half of the unfavorable prognostic characteristics were included in the poor prognosis group. This scoring system was used to stratify participants and analyze the relationship between prognostic scores and clinical outcomes.

### 2.4. Statistical Analysis

Patient data collected during the study were analyzed using the IBM Statistical Package for the Social Sciences (SPSS) for macOS, version 29.0 (IBM Corp., Armonk, NY, USA). Categorical variables were summarized as frequencies and percentages, while continuous variables were described using means, standard deviations, medians, and minimum–maximum values. The normality of the data was assessed using the Kolmogorov–Smirnov test. Relationships between continuous variables were analyzed using Spearman’s correlation analysis. Receiver operating characteristic (ROC) curve analysis was conducted to evaluate the discriminatory power of the walking tests in predicting poor prognosis. ROC curves were generated to determine optimal cut-off values, sensitivity, specificity, and other related metrics. Statistical significance was defined as a *p*-value of less than 0.05.

## 3. Results

The study included 129 participants, 76% of whom were female and 24% male, with a mean age of 38 years (range: 19–66). The majority of participants (86.8%) were diagnosed with relapsing–remitting multiple sclerosis (RRMS), while 7.8% had primary progressive MS (PPMS), and 5.4% had secondary progressive MS (SPMS). The demographic and clinical characteristics are detailed in Table 1.

Analysis of correlations between the 25-foot walk tests and other clinical measures revealed statistically significant findings. Forward walking times showed a moderate positive correlation with current EDSS scores (r = 0.526, *p* < 0.001) and a strong positive correlation with the 9HPT for the dominant hand (r = 0.609, *p* < 0.001). Similarly, backward walking times were moderately correlated with EDSS (r = 0.469, *p* < 0.001) and exhibited weaker but still significant correlations with the 9HPT for the dominant hand (r = 0.452, *p* < 0.001). Both forward and backward walking times were negatively correlated with SDMT scores (r = −0.427, *p* < 0.001 for both), indicating that slower walking speeds are associated with reduced cognitive function. Additionally, backward walking times demonstrated a weak but significant positive correlation with disease duration (r = 0.245, *p* = 0.005), as shown in Table 2.

The utility of 25-foot walk tests in identifying poor prognosis was further explored through ROC curve analysis. The forward walking test displayed high discriminatory power, with an area under the curve (AUC) of 0.857 (95% CI: 0.739–0.975) and a cut-off time of >5.7 s (sensitivity: 88.2%, specificity: 75.0%) (Figure 1) (Table 3). Backward walking also demonstrated strong discriminative ability, achieving an AUC of 0.832 (95% CI: 0.705–0.959) with a cut-off time of >11.11 s (sensitivity: 76.5%, specificity: 84.8%).

Prognostic analyses indicated that the forward walking test achieved an AUC of 0.703 (95% CI: 0.587–0.818) with a threshold of >5.5 s, while the backward walking test had an AUC of 0.670 (95% CI: 0.547–0.792) with a threshold of >11.8 s (Figure 2). Although both tests were significant predictors, the forward walking test demonstrated slightly higher sensitivity and specificity for prognostic purposes.

## 4. Discussion

This study evaluated the clinical utility and sensitivity of the B-T25FW in individuals with MS. Our findings demonstrate that backward walking is a practical, clinically meaningful measure that sensitively detects mobility impairments, showing significant correlations with established clinical outcomes, including the EDSS, SDMT, 9HPT, and disease duration. Additionally, prognostic analyses revealed that the forward 25-foot walk test demonstrated higher sensitivity and specificity, indicating differences in their utility for predicting disease outcomes.

Backward walking inherently poses greater challenges than forward walking, requiring higher levels of neuromuscular coordination, postural control, and sensory feedback. These demands engage multiple neurophysiological systems that are commonly affected in MS. In particular, executing a backward gait places increased reliance on descending motor pathways (corticospinal tracts) for precise limb control, cerebellar circuits for coordinating movement and maintaining balance, and proprioceptive feedback systems for spatial orientation in the absence of visual cues [18,19,20]. Dysfunction in any of these pathways—due to MS-related demyelination and axonal loss—can disproportionately impair backward walking performance, making the B-T25FW especially sensitive to subtle deficits that might go undetected during forward gait. Consequently, even mild impairments in balance or limb position sense can significantly slow backward walking, a phenomenon observed in MS, in which subclinical motor and sensory deficits are common [9,18]. This heightened sensitivity was evident in our findings: B-T25FW times correlated significantly with the EDSS and the 9HPT performance of the dominant hand. The moderate correlation with EDSS underscores that backward walking reflects global disability, as EDSS scoring incorporates motor (particularly pyramidal tract) and sensory components critical to MS progression. This aligns with prior studies such as that of Söke et al. [10], which reported moderate-to-strong associations between backward walking performance and EDSS, highlighting the ability of backward gait tests to capture overall disability in MS. Similarly, Edwards et al. [8] demonstrated that backward walking velocity distinguishes fallers from non-fallers, further emphasizing its sensitivity to functional impairments.

Similarly, the association with the 9HPT highlights the interconnectedness of lower and upper extremity motor functions in MS. Studies such as Benedict et al. [13] and Feys et al. [14] have established the 9HPT as a robust measure of manual dexterity that correlates with global motor disability, including walking impairments. Our findings suggest that backward walking and upper limb function share neuromuscular and proprioceptive control mechanisms, making both assessments valuable for capturing comprehensive motor impairments. Additionally, the B-T25FW’s reliance on dynamic postural adjustments and sensory-motor integration mirrors the demands of the 9HPT, further supporting their interrelationship. This connection may also be attributed to shared neural networks involved in coordinating complex motor tasks, as suggested by Rao et al. [21]. However, compared to backward walking, the 9HPT provides a more targeted evaluation of fine motor skills, while the B-T25FW integrates broader neuromuscular systems, including balance and proprioception. Despite these strengths, the slightly lower correlation between the B-T25FW and the 9HPT compared to EDSS suggests that backward walking captures unique aspects of functional impairment not fully represented by upper limb assessments. This highlights the complementary roles of these tests in providing a multidimensional view of disability in MS. Future studies should explore combining the B-T25FW and 9HPT to improve sensitivity in detecting subtle deficits, particularly in early-stage MS. Advanced technologies like wearable sensors could further refine these assessments, enhancing precision and tracking disease progression.

Furthermore, the cognitive demands of backward walking, particularly in dual-task scenarios, provide valuable insights into the interplay between motor and cognitive systems in MS. Our findings revealed a significant negative correlation between SDMT scores and backward walking performance, consistent with prior studies emphasizing the role of information processing speed in mobility impairments in MS [13,22]. Impaired cognitive-motor integration, as reflected by slower SDMT scores, has been linked to deficits in dynamic balance and gait control, further substantiating backward walking as a sensitive tool for assessing cognitive and motor interplay [18,21]. This association is particularly relevant in the context of MS, where subtle cognitive impairments often manifest as mobility deficits, even in patients with relatively low physical disability scores. For example, Johansson et al. [23] highlighted that tasks requiring simultaneous motor and cognitive engagement, such as backward walking, are more sensitive to early functional impairments. By integrating backward walking assessments with cognitive tests like the SDMT, clinicians can gain a comprehensive understanding of both physical and cognitive dimensions of disability, enabling more targeted interventions and monitoring strategies.

Our findings reinforce the utility of backward walking assessments in clinical practice. While the B-T25FW demonstrated sensitivity and specificity for predicting mobility impairments, the forward T25FW exhibited slightly superior prognostic accuracy for poor outcomes. This may reflect the alignment of forward walking tasks with real-world mobility demands and their more established clinical thresholds [24]. However, backward walking remains highly sensitive to subtle deficits, requiring greater neuromuscular coordination and postural control [18,23]. Despite its slightly lower prognostic power, the B-T25FW complements forward walking tests by detecting early impairments, particularly in balance and cognitive-motor integration [8,10]. Future research should focus on validating cut-off scores for the B-T25FW and exploring its integration with wearable technologies to enhance precision and applicability. Together, forward and backward walking assessments could provide a more comprehensive evaluation of functional impairment and fall risk in MS. Backward walking also offers potential as a rehabilitation target. Recent evidence suggests that training programs emphasizing backward walking can improve postural control, strength, and balance, thereby reducing fall risk [10]. The development of standardized protocols for backward walking interventions could enhance their clinical utility, particularly for patients with high fall risk.

While our study provides robust evidence supporting the clinical utility of the B-T25FW, several limitations warrant consideration. First, the cross-sectional design precludes conclusions about the longitudinal predictive value of the B-T25FW. Future longitudinal studies are needed to establish its utility in tracking disease progression and fall risk over time. Additionally, integrating backward walking with other mobility tests and neuroimaging could provide a more comprehensive understanding of functional impairments in MS [25]. Second, the high proportion of participants with RRMS may limit the generalizability of findings to progressive MS subtypes, which often exhibit greater motor impairments and may demonstrate different patterns of backward walking performance [26]. Third, although cognitive measures like the SDMT were included, other domains such as executive function and memory were not explicitly evaluated. These cognitive domains influence walking performance and fall risk in MS and should be addressed in future studies [13]. Finally, this study did not use advanced motion-capture or wearable technologies for assessing backward walking. Such tools could enhance the precision of measurements and capture subtle impairments not detectable with manual methods [27]. Despite its limitations, this study has several notable strengths. The study addresses a critical gap by demonstrating the potential of backward walking as a marker of both mobility and cognitive-motor integration. By incorporating validated measures such as the EDSS, SDMT, and NHPT, it provides strong evidence for the multidimensional nature of backward walking impairments in MS. Additionally, the detailed evaluation of backward walking as a complement to forward walking highlights their distinct roles, with backward walking offering unique insights into balance, proprioception, and postural control [18]. Finally, the inclusion of diverse clinical and demographic variables, such as fatigue and disease duration, allows for a more comprehensive understanding of factors influencing backward walking performance.

## 5. Conclusions

Our findings support the B-T25FW as a clinically useful and sensitive assessment for mobility impairment in individuals with MS. By highlighting its correlations with key clinical measures such as the EDSS, SDMT, and NHPT, we establish backward walking as a valuable complementary metric to traditional forward walking assessments. The B-T25FW uniquely captures aspects of balance, proprioception, and cognitive-motor integration, providing multidimensional insights into functional impairments associated with MS. Future research should focus on validating these findings in diverse MS populations, exploring its utility in longitudinal disease tracking and optimizing its use in rehabilitation strategies.

## Figures and Tables

**Figure 1 diagnostics-15-00936-f001:**
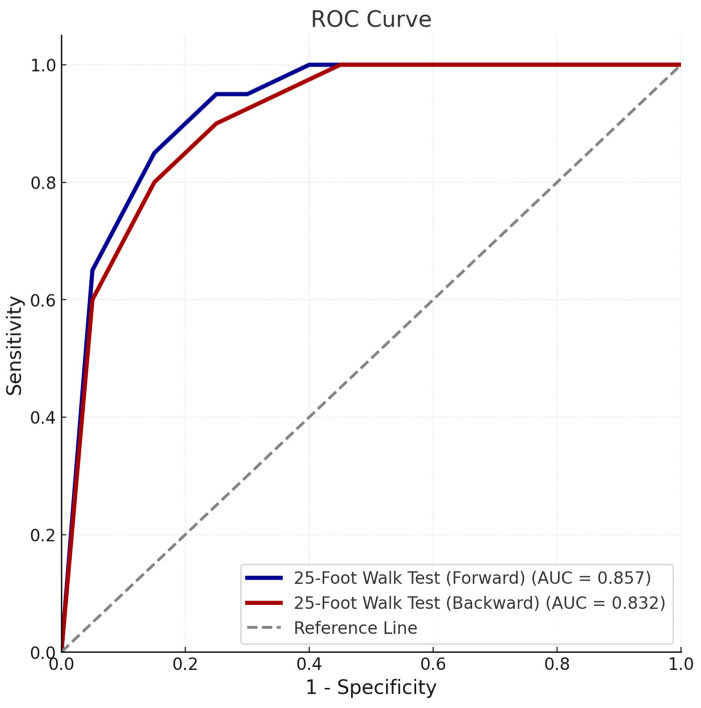
Receiver operating characteristic (ROC) curve demonstrating the discriminatory ability of forward and backward 25-foot walk tests for poor prognosis based on clinical form. The area under the curve (AUC), cut-off values, sensitivity, and specificity for each test are shown.

**Figure 2 diagnostics-15-00936-f002:**
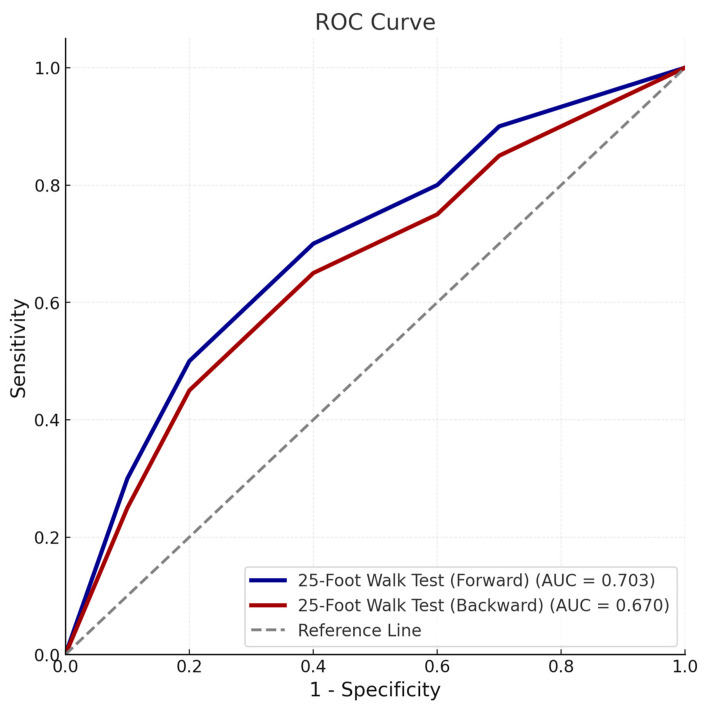
Receiver operating characteristic (ROC) curve illustrating the predictive accuracy of forward and backward 25-foot walk tests for poor prognosis based on the prognostic scoring system. The area under the curve (AUC), cut-off values, sensitivity, and specificity are detailed for each parameter.

**Table 1 diagnostics-15-00936-t001:** Demographic, clinical, functional, and imaging characteristics of participants.

Variable	Mean ± SD	Median (Min–Max)	*n* (%)
Sex			
Female			98 (76)
Male			31 (24)
Age at last visit (years)	38 ± 11	38 (19–66)	
BMI (kg/m^2^)	24.6 ± 4.5	24 (15.4–41.6)	
Current smoker			44 (34.1)
Former smoker			14 (10.9)
Never			71 (55)
MS Disease Duration (years)	7.4 ± 6	6 (1–25)	
Age at First Attack (years)	29.1 ± 9.1	27 (13–55)	
RRMS			112 (86.8)
PPMS			10 (7.8)
SPMS			7 (5.4)
Initial MS Presentation			
Polysymptomatic			54 (41.9)
Monosymptomatic			75 (58.1)
Treatment Type			
Low Efficacy			58 (44.9)
High Efficacy			69 (53.4)
Total Number of Attacks	2.3 ± 1.7	2 (0–8)	
EDSS at Diagnosis	1.9 ± 1.3	1 (0–6)	
Current EDSS	2 ± 1.4	1.5 (0–6)	
Comorbidities			30 (23.3)
9HPT (Dominant Hand) (s)	19.9 ± 5.6	18.8 (5.4–53.7)	
9HPT (Non-Dominant Hand)(s)	21.6 ± 6.4	20.5 (10.8–67.5)	
SDMT Correct Responses	40.4 ± 14.4	42 (5–77)	
Forward T25FW (s)	5.9 ± 3	5.2 (3.7–32.5)	
Backward T25FW (s)	10.4 ± 6.5	8.6 (4.4–50)	
MS Quality of Life Score	37.8 ± 23.1	30 (0–100)	
VAS Fatigue Score	4.8 ± 2.4	5 (0–10)	
Prognostic Score	3.2 ± 1.5	3 (0–7)	
Good Prognosis			107 (82.9)
Poor Prognosis			22 (17.1)
T2 Lesion Count (Initial MRI)	11.5 ± 5.4	10 (2–30)	
Gadolinium-Enhancing Lesions			43 (33.3)
Presence of Cervical Lesions			51 (39.5)
Upper Cervical Lesions (C1–C4)			41 (46.1)

Abbreviations: BMI: body mass index, EDSS: Expanded Disability Status Scale, RRMS: relapsing–remitting multiple sclerosis, PPMS: primary progressive multiple sclerosis, SPMS: secondary progressive multiple sclerosis, 9HPT: Nine-Hole Peg Test, SDMT: Symbol Digit Modalities Test, T25FW: Timed 25-Foot Walk Test, VAS: visual analog scale, MRI: magnetic resonance imaging.

**Table 2 diagnostics-15-00936-t002:** Relationship between 25-foot walk test results and clinical features.

Clinical Feature		Forward Walking Test	Backward Walking Test
EDSS at Diagnosis	r	0.407	0.345
	*p*	<0.001	<0.001
Current EDSS	r	0.526	0.469
	*p*	<0.001	<0.001
Nine-Hole Peg Test (Dominant Hand)	r	0.609	0.452
	*p*	<0.001	<0.001
Nine-Hole Peg Test (Non-Dominant Hand)	r	0.568	0.410
	*p*	<0.001	<0.001
SDMT Correct Responses	r	−0.427	−0.427
	*p*	<0.001	<0.001
SDMT Total Responses	r	−0.425	−0.425
	*p*	<0.001	<0.001
MS Quality of Life Score	r	0.404	0.337
	*p*	<0.001	<0.001
VAS Fatigue Score	r	0.390	0.286
	*p*	<0.001	0.001
Prognostic Score	r	0.298	0.117
	*p*	0.001	0.186
Disease Duration	r	0.153	0.245
	*p*	0.084	0.005

The last two columns indicate correlation coefficients (r) and their respective *p*-values. Abbreviations: EDSS: Expanded Disability Status Scale, SDMT: Symbol Digit Modalities Test, VAS: visual analog scale, r: correlation coefficient.

**Table 3 diagnostics-15-00936-t003:** ROC analysis of poor prognosis for 25-foot walk tests.

Risk Factor	AUC (95% CI)	Cut-Off	*p*-Value	Sensitivity (%)	Specificity (%)	PPV (%)	NPV (%)
Clinic Form							
25-Foot Walk (Forward)	0.857 (0.739–0.975)	>5.70	<0.001	88.2	75	34.9	97.7
25-Foot Walk (Backward)	0.832 (0.705–0.959)	>11.11	<0.001	76.5	84.8	43.3	96
Prognosis Measurement							
25-Foot Walk (Forward)	0.703 (0.587–0.818)	>5.50	0.001	68.2	68.2	30.6	91.3
25-Foot Walk (Backward)	0.670 (0.547–0.792)	>11.80	0.007	45.5	87.9	43.5	88.7

Abbreviations: AUC: area under curve, PPV: positive predictive value, NPV: negative predictive value.

## Data Availability

The raw data supporting the conclusions of this article will be made available by the authors on request.

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
