# Peer review of "Backward Walking as a Marker of Mobility and Disability in Multiple Sclerosis: A Cross-Sectional Analysis"

_diagnostics, 2025, doi:10.3390/diagnostics15070936_

Round 1

Reviewer 1 Report

Comments and Suggestions for Authors

In this cross-sectional study, the authors examined the feasibility and sensitivity of the backward Timed 25-Foot Walk Test (B-T25FW) in 129 MS patients. Disability (EDSS), cognition (SDMT), manual dexterity (9HPT), fatigue, and both forward/backward walking times were assessed. Both walking measures correlated with key clinical metrics, including a moderate correlation between B-T25FW and EDSS. While forward walking had slightly stronger predictive power, the authors conclude that B-T25FW is a viable, sensitive measure of mobility in MS and should be further validated longitudinally.

The introduction states the aim is to evaluate the “feasibility and sensitivity” of the backward T25FW. Clarifying how these terms are operationally defined (e.g., what constitutes “feasibility” in this context—time to administer, patient tolerance, safety, etc.?) would strengthen the rationale.

A power analysis was performed prior to the study; however, please also calculate the achieved power given the final sample size of 129 participants.

Lines 121–122 (“The B-T25FW has been shown to provide unique insights into balance and mobility challenges in MS patients [Fritz et al., 2013].”) seem more like background/contextual information and disrupts the flow of the current section. Consider moving this sentence to the Introduction.

The criteria for determining “favorable” vs. “unfavorable” prognosis are unclear. It appears that all parameters listed in lines 129–138 refer to unfavorable prognosis. Please revise the beginning of this section to read: “The parameters for unfavorable prognosis included:” (line 129).

Please describe low- and high-efficacy treatment types in the Methods section.

In Table 2, clarify that the last two columns represent p-values.

The quality of Figures 1 and 2 should be improved for better readability.

Author Response

Reviewer 1

Comments and Suggestions for Authors

In this cross-sectional study, the authors examined the feasibility and sensitivity of the backward Timed 25-Foot Walk Test (B-T25FW) in 129 MS patients. Disability (EDSS), cognition (SDMT), manual dexterity (9HPT), fatigue, and both forward/backward walking times were assessed. Both walking measures correlated with key clinical metrics, including a moderate correlation between B-T25FW and EDSS. While forward walking had slightly stronger predictive power, the authors conclude that B-T25FW is a viable, sensitive measure of mobility in MS and should be further validated longitudinally.

  1. The introduction states the aim is to evaluate the “feasibility and sensitivity” of the backward T25FW. Clarifying how these terms are operationally defined (e.g., what constitutes “feasibility” in this context—time to administer, patient tolerance, safety, etc.?) would strengthen the rationale.

We thank the reviewer for highlighting the potential ambiguity in the term 'feasibility.' To avoid speculation and clearly represent our study's purpose, we have revised our aim statement. We now explicitly emphasize the clinical relevance, practical applicability, and sensitivity of the B-T25FW by clearly indicating our focus on its correlations with established clinical measures. Similar changes were done in abstract, discussion and conclusion sections.

  1. A power analysis was performed prior to the study; however, please also calculate the achieved power given the final sample size of 129 participants.

Thanks fort his comment. Using the final sample size n = 129, we recalculate with δ ≈ 4.54 as above. The resulting power is dramatically higher due to the larger sample. In fact, the achieved power is approximately 99% (around 0.995 or 99.5% by calculation). This means that the test is extremely well-powered: it will correctly reject the null hypothesis about 99 times out of 100 under the given alternative.

  1. Lines 121–122 (“The B-T25FW has been shown to provide unique insights into balance and mobility challenges in MS patients [Fritz et al., 2013].”) seem more like background/contextual information and disrupts the flow of the current section. Consider moving this sentence to the Introduction.

Thank you for highlighting this redundancy. We have revised the paragraph in the introduction to remove repetition and enhance clarity.

  1. The criteria for determining “favorable” vs. “unfavorable” prognosis are unclear. It appears that all parameters listed in lines 129–138 refer to unfavorable prognosis. Please revise the beginning of this section to read: “The parameters for unfavorable prognosis included:” (line 129).

Agree. The suggested revision was done.

  1. Please describe low- and high-efficacy treatment types in the Methods section.

We thank you for emphasizing this important clarification. As you recommended, we have explicitly defined low- and high-efficacy treatment categories in the Procedures subsection.

  1. In Table 2, clarify that the last two columns represent p-values.

We thank the you for pointing out this potential ambiguity. We have revised the relevant section.

  1. The quality of Figures 1 and 2 should be improved for better readability.

Agree. Both figures were redrawn more clearly and distinctly.

Reviewer 2 Report

Comments and Suggestions for Authors

Dear Authors,

The study aims to evaluate the feasibility and sensitivity of the Backward Timed 25-Foot Walk Test (B-T25FW) as a clinical tool in Multiple sclerosis (MS) by exploring its associations with various demographic, clinical, functional, and cognitive measures in a well-characterized cohort. Assessing gait deficits in MS is important because they increase the risk of falls and contribute to activity limitation, social isolation, and decreased quality of life. The problem has been covered in many publications. Several reliable tests are usually used to assess motor deficits. The Authors proposed to use of Backward Timed 25-Foot Walk Test (B-T25FW) as a clinical tool in Multiple sclerosis because it requires more active cognitive and proprioceptive engagement.

The study design is clear. The findings suggest changes in both the forward and backward 25-foot walk tests for poor prognosis in multiple sclerosis. However, the backward 25-foot walk test is characterized by lower sensitivity. Therefore, it is necessary to clarify the advantage of using the backward 25-foot walk test in patients with MS. The discussion needs to be modified to clarify how the backward 25-foot walk test reflects the impairment of neurophysiological mechanisms of motor control in MS. A deeper understanding of the obtained results is required.

The maintext should be formatted according to the guidelines. Figure 1, 2 are not mentioned in the text.

Author Response

Dear Authors,

The study aims to evaluate the feasibility and sensitivity of the Backward Timed 25-Foot Walk Test (B-T25FW) as a clinical tool in Multiple sclerosis (MS) by exploring its associations with various demographic, clinical, functional, and cognitive measures in a well-characterized cohort. Assessing gait deficits in MS is important because they increase the risk of falls and contribute to activity limitation, social isolation, and decreased quality of life. The problem has been covered in many publications. Several reliable tests are usually used to assess motor deficits. The Authors proposed to use of Backward Timed 25-Foot Walk Test (B-T25FW) as a clinical tool in Multiple sclerosis because it requires more active cognitive and proprioceptive engagement.

  1. The study design is clear. The findings suggest changes in both the forward and backward 25-foot walk tests for poor prognosis in multiple sclerosis. However, the backward 25-foot walk test is characterized by lower sensitivity. Therefore, it is necessary to clarify the advantage of using the backward 25-foot walk test in patients with MS. The discussion needs to be modified to clarify how the backward 25-foot walk test reflects the impairment of neurophysiological mechanisms of motor control in MS. A deeper understanding of the obtained results is required.

We appreciate this insightful recommendation. In line with your suggestion, we've expanded the Discussion section to explicitly include neurophysiological mechanisms, such as corticospinal tract dysfunction, cerebellar involvement, and impaired proprioceptive feedback, to better clarify how backward walking uniquely reflects motor impairment in MS. 

  1. The maintext should be formatted according to the guidelines. Figure 1, 2 are not mentioned in the text.

Thank you for highlighting this formatting oversight. We have carefully reformatted the manuscript to fully align with the journal guidelines. Additionally, Figures 1 and 2 are now clearly referenced within the main text, and their positions have been appropriately rearranged for improved clarity and coherence.

Round 2

Reviewer 2 Report

Comments and Suggestions for Authors

The manuscript has been sufficiently improved to warrant publication in Diagnostics. All comments have been considered.